# Optimization of Oligomer Chitosan/Polyvinylpyrrolidone Coating for Enhancing Antibacterial, Hemostatic Effects and Biocompatibility of Nanofibrous Wound Dressing

**DOI:** 10.3390/polym14173541

**Published:** 2022-08-29

**Authors:** Vinh Khanh Doan, Chien Minh Tran, Trinh Thi-Phuong Ho, Linh Kim-Khanh Nguyen, Yen Ngoc Nguyen, Ngan Tuan Tang, Tin Dai Luong, Nhi Ngoc-Thao Dang, Nam Minh-Phuong Tran, Binh Thanh Vu, Hoai Thi-Thu Nguyen, Quyen Thuc Huynh, Hien Quoc Nguyen, Chien Mau Dang, Thang Bach Phan, Hanh Thi-Kieu Ta, Viet Hung Pham, Thanh Dinh Le, Toi Van Vo, Hiep Thi Nguyen

**Affiliations:** 1Tissue Engineering and Regenerative Medicine Laboratory, Department of Tissue Engineering and Regenerative Medicine, School of Biomedical Engineering, International University, Ho Chi Minh City 700000, Vietnam; 2Vietnam National University, Ho Chi Minh City 700000, Vietnam; 3School of Biotechnology, International University, Ho Chi Minh City 700000, Vietnam; 4Vietnam Atomic Energy Institute, 59 Ly Thuong Kiet, Hanoi 100000, Vietnam; 5Institute for Nanotechnology, Ho Chi Minh City 700000, Vietnam; 6Center for Innovative Materials and Architectures (INORMAR), Ho Chi Minh City 700000, Vietnam; 7Laboratory of Advanced Materials, University of Science, Ho Chi Minh City 700000, Vietnam; 8Faculty of Materials Science and Technology, University of Science, Ho Chi Minh City 700000, Vietnam; 9Key Laboratory of Analytical Technology for Environmental Quality and Food Safety Control, University of Science, Vietnam National University, Hanoi 100000, Vietnam; 10Thong Nhat Hospital, Ho Chi Minh City 700000, Vietnam

**Keywords:** oligomer chitosan (COS), polyvinylpyrrolidone (PVP), polycaprolactone (PCL), hemostatic effect, antibacterial wound dressing

## Abstract

A synergistic multilayer membrane design is necessary to satisfy a multitude of requirements of an ideal wound dressing. In this study, trilayer dressings with asymmetric wettability, composed of electrospun polycaprolactone (PCL) base membranes coated with oligomer chitosan (COS) in various concentrations of polyvinylpyrrolidone (PVP), are fabricated for wound dressing application. The membranes are expected to synergize the hygroscopic, antibacterial, hemostatic, and biocompatible properties of PCL and COS. The wound dressing was coated by spraying the solution of 3% COS and 6% PVP on the PCL base membrane (PVP6–3) three times, which shows good interaction with biological subjects, including bacterial strains and blood components. PVP6–3 samples confirm the diameter of inhibition zones of 20.0 ± 2.5 and 17.9 ± 2.5 mm against *Pseudomonas aeruginosa* and *Staphylococcus aureus*, respectively. The membrane induces hemostasis with a blood clotting index of 74% after 5 min of contact. In the mice model, wounds treated with PVP6–3 closed 95% of the area after 10 days. Histological study determines the progression of skin regeneration with the construction of granulation tissue, new vascular systems, and hair follicles. Furthermore, the newly-growth skin shares structural resemblances to that of native tissue. This study suggests a simple approach to a multi-purpose wound dressing for clinical treatment.

## 1. Introduction

As our first defensive line, the skin performs various roles, from protecting against harmful factors of the outer environment to regulating and maintaining the inner environmental condition, as well as connecting with the nervous system for sensation. Protective functions of the skin can be disrupted by various types of damage, which might lead to pathogen invasion and severe infections demanding tremendous effort in treatment. As a result, wound dressings are critically needed for clinical treatment, especially for severe cases such as burn wounds with a large surface area and chronic wounds due to diabetes or infection [1,2]. To effectively improve wound treatment, wound dressings must satisfy several fundamental requirements, including reliable mechanical strength, high absorbability of body fluid and exudate, induction of blood coagulation, ability to balance environmental moisture, prevention of dust and bacterial invasion, as well as antibacterial activity against several bacterial strains [3].

A potential design to fabricate such an ideal wound dressing is a mechanically strong basement membrane coated with a layer of bioagents that accelerates the wound healing process and prevents biofilm formation [4]. Synthetic biopolymers such as poly (L-lactic acid) (PLLA) [5], poly (lactic acid-co-glycolic acid) (PLGA) [6], and polycaprolactone (PCL) [7] are favorable for the fabrication of the base layer thanks to their inherently strong mechanical properties. Among these, PCL has been acknowledged as a promising candidate that exhibits both mechanical strength and biocompatibility in several applications [8]. The combination of the PCL and electrospinning techniques has been employed to create a fibrous scaffold with an interconnected porous structure that stimulates the formation of natural extracellular matrix (ECM) [7]. Furthermore, the electrospun PCL membrane is highly adaptable and effectively prevents water infiltration thanks to its mechanical strength and hydrophobicity [9]. Despite not sharing the biochemical signatures of native tissues, hybrid biomaterials between PCL and natural polymers can be engineered via many approaches to retain the mechanical strength and durability of PCL and the biological functionality of natural polymers [10].

Since the hydrophobicity of PCL hindered its ability to hold a coating layer of bioactive agents, surface modification techniques were employed to improve its wettability, especially plasma treatment that functionalizes the membrane surface with hydrophilic functional groups [11]. Unfortunately, the introduced moieties had low stability and a short lifespan [12]. The alternative method of polymer blends prior to electrospinning has been proposed to take advantage of the hydrophilicity of the added material [13]. Compared to common hydrophilic polymers, such as polyvinyl alcohol and polyethylene glycol, amphiphilic polymers with lower surface energy showed better segregation on the surface and thus rendered more desirable wettability [14]. The group of surfactants from tri-block copolymer, especially poloxamer (POX), consists of two hydrophilic side-chains of polyethylene oxide and one middle hydrophobic chain of polypropylene oxide, which can be used as an additional blending material for wettability modulation [15,16]. In addition, previous reports have reported that surfactants such as POX could restrict bacterial adhesion and thus reduce biofilm formation by changing cellular membrane characteristics [17,18].

Among different bioactive agents for coating purposes, such as plant or herb extractions and essential oils, chitosan oligosaccharide (COS), derived from chitin of abundant crustacean sources, has attracted great attention for its antimicrobial and hemostatic activities [19]. However, a coating layer of only COS is brittle because of its low molecular weight, which decreases the overall mechanical strength of the membrane. COS is highly water-soluble and tends to dissolve into the environment rather than exhibit water absorption behavior. Therefore, the combination of COS with a synthetic polymer such as polyvinylpyrrolidone (PVP) could synergize their properties. PVP is a nonionic and water-soluble polymer that was approved by the FDA [20] for pharmaceutical applications such as film-forming agents, binders, stabilizers, or material for nanoparticle synthesis [21,22,23]. Known for its mechanical stability, swelling behavior, and low cytotoxicity, many researchers have blended COS and PVP to produce many new composites in the forms of hydrogel [24], sponge [25], membrane [26], or film [27] to compensate for COS limitation. Nevertheless, most studies focused on enhancing the antibacterial activity, hemostasis, and wound regeneration by adding other additives, such as graphene oxide [28] and bentonite [29], instead of utilizing the synergistic effect of PVP and COS itself. Since the PVP:COS ratios were often fixed, the PVP influence on the integral properties was neglected as well.

Herein, this research demonstrated a simple strategy to coat bioactive PVP/COS onto an electrospun PCL membrane through surface modification with POX. The coating step was performed by spraying the COS and PVP mixture onto the hydrophilic PCL/POX layer of the bilayer PCL-PCL/POX membrane. The effects of PVP:COS ratios and spraying time on the membrane’s properties were investigated. The formation of PVP/COS on the PCL/POX surface was confirmed by SEM and FTIR methods. The PCL-PCL/POX-PVP/COS membranes possessed good tensile strength and fluid absorbability. Moreover, the synergistic effect of PVP and COS is expected to impart antibacterial and hemostatic activities to the membrane. The fabrication method and application of PCL-PCL/POX-PVP/COS membrane were illustrated in Figure 1.

## 2. Materials & Methods

### 2.1. Materials

Poly (ɛ-caprolactone) (PCL, Mn 80.000) and Poloxamer 407 (POX) were obtained from Sigma-Aldrich Co. (St. Louis, MO, USA). Polyvinylpyrrolidone (PVP K30, Mn 40,000), acetone (AC, CH_3_COCH_3_, 99.5%), Hydrogen Peroxide (H_2_O_2_) 30% (*w*/*w*) reagent, and EtOH (C_2_H_5_OH) (100%) were purchased from Xilong Chemical Co., Ltd. (Guangdong, China). Low viscosity chitosan (CS) from shrimp shells was purchased from the Dao Nguyen company, Vietnam. The pathogens, including Staphylococcus aureus (SA) ATCC 25,913 and Pseudomonas aeruginosa (PA) ATCC 9028, were provided by the Marine Laboratory, International University-HCM, Vietnam National University (Vietnam). The Mueller Hinton Broth (M391-500G) was purchased from Hi-Media (Maharashtra, India).

### 2.2. Methodology

#### 2.2.1. Fabrication of PCL and PCL/POX Bilayer Membrane

The electrospun membrane was prepared based on the reported protocol [9]. Briefly, a 45 mL PCL solution (15% *w*/*v*) was prepared by dissolving 6.75 g PCL pellets in 45 mL of acetone. Similarly, a 15 mL PCL/POX solution was prepared by dissolving 2.25 g PCL and 0.19 g POX (a ratio of 12:1 (*w*/*w*)) in 15 mL of acetone. The mixtures were stirred overnight at 50 °C. Prepared PCL and PCL/POX solutions were filled into syringes connected to a 20G needle. The electrospinning parameters were set up as follows: feed-rate of 1 mL/h, voltage of 15 kV, tip-to-collector distance of 10 cm, and rotating speed of 150 rpm. Those two solutions, PCL and PCL/POX, were electrospun in sequence to attain a bilayer membrane. The electrospun membrane was later detached from the collector for further investigation.

#### 2.2.2. Preparation of PVP/COS Solution

The preparation of 100 mL of COS solution (3%) was performed according to the reported protocol with some modifications [30]. Briefly, 3 g of chitosan powder was immersed in the 100 mL H_2_O_2_ solution (6%) at 30 °C for 10 min. Then, the mixture was irradiated at 400 W in the microwave oven for a total of three minutes. After every 1 min, the mixture was taken outside for 30 s to reduce the temperature and slow down the water evaporation.

#### 2.2.3. Fabrication of PCL-PCL/POX-PVP/COS Membranes

The membranes were cut into square pieces of 50 × 50 mm^2^. Next, PVP/COS solutions with various PVP concentrations were poured into the chamber of the hand spray gun, Total TT3506 500W (Total Tools Co., PTE. LTD, Jiangsu, China). The working parameters were set at 200 mL/min spraying speed with 2 s per spray, and the gun was fixed horizontally at 40 cm above the PCL-PCL/POX membranes. Then, each solution was sprayed on PCL-PCL/POX layers 3 or 6 times. During the spraying process, the membrane was placed in the oven at 50 °C for 30 min after each spraying time for dehydration. Finally, all the samples were vacuum dried at 50 °C until completely dried. The fabrication parameters of the samples tested in this study are summarized in Table 1 below.

#### 2.2.4. Physicochemical Characterization of PCL-PCL/POX-PVP/COS Membrane

The surface morphology of the membranes was observed using SEM (JSM-IT100, JEOL, Tokyo, Japan) after they were sputter-coated with gold (JEOL Smart Coater, Tokyo, Japan) at 10 kV. A total of 30 fibers and 30 pores were chosen randomly from SEM micrographs and their diameters were measured by ImageJ software (NIH, Maryland, USA).

Membrane hydrophilicity was determined by measuring the surface’s water contact angle. The membranes were completely dried prior to measurement to avoid water interaction issues. A 5 µL deionized water (DI) droplet was dropped from a micropipette onto the surface of the membranes. The water contact angle (WCA) was captured by a DSLR camera (Canon, Tokyo, Japan) and measured by ImageJ software. The experiment was replicated four times.

The presence of chemical groups in the samples was determined by using Fourier-transform Infrared Spectroscopy (FT-IR, Spectrum GX, PerkinElmer Inc., Waltham, MA, USA).

Water absorption was tested according to the BS EN 13726-1:2002 standard. Briefly, the fully-dehydrated membranes were cut into pieces the size of 50 × 50 mm^2^ and placed on Petri dishes. Then, a sufficient amount of test solution was poured into the dishes at a weight ratio of 1:40 and incubated at 37 °C for 30 min. The test solution was comparable with body serum and wound exudate, containing 142 mmol Na^+^ and 2.5 mmol Ca^2+^ in chloride salt form. Afterward, the membranes were gently shaken to remove unabsorbed water and weighed using an analytical scale. The water uptake of samples was calculated by the following equation:(1)$$\mathrm{Water}\;\mathrm{uptake}=\frac{W_{f}-W_{i}}{25}\;\left(\mathrm{mg}/{\mathrm{cm}}^{2}\right)$$
where W_i_ is the initial weight of the membrane and W_f_ is the weight of the sample after absorbing the artificial exudate solution. The experiment was replicated ten times.

The mechanical strength of membranes was evaluated by tensile test using the Texture Analyser (TA.XTplus, Stable Micro Systems, Surrey, UK). Samples were cut into 40 × 10 mm^2^ rectangles and vertically held by two mechanical miniature grips, leaving 30-mm-length for tensile loading. The loading speed of the experiment was fixed at 8.3 mm/s. The sample thickness was predetermined using an electronic micrometer.

#### 2.2.5. In Vitro Biological Characterization of PCL-PCL/POX-PVP/COS Membrane

##### Antibacterial Activity of PCL-PCL/POX-PVP/COS Membrane

The antimicrobial fabric zone of inhibition test (AATCC 147) was applied to evaluate the antibacterial effects of the PVP/COS layer on two bacterial strains, including *Staphylococcus aureus* (SA) and *Pseudomonas aeruginosa* (PA). Prior to the experiments, a colony of each strain was collected from a stock agar plate, transferred into a 5 mL MHB tube, and cultured at 37 °C for 24 h. Then the bacterial suspension of each strain was diluted to acquire an optical density at 620 nm OD620 = 0.08−0.1 (equal to 0.5 McFarland standards, approximately 1−2 × 10^8^ CFU/mL).

The microbial susceptibility of each sample against a specific bacterial strain was examined separately on the Mueller–Hinton agar (MHA) plate. Briefly, 150 µL of the diluted bacterial suspension was dropped into and spread on the MHA surface. Then, square membranes with a size of 10 × 10 mm^2^ were placed on the MHA plate and incubated for 24 h. The pure PCL electrospun membrane and the AQUACEL^®^ Ag Extra^TM^ wound dressing were chosen as the negative and positive controls, respectively. The bacterial inhibitory zone around the samples was measured after incubation.

##### Assessment of In Vitro Blood Coagulation

The in vitro blood coagulation research was approved by the Institutional Review Board of the School of Biomedical Engineering, Ho Chi Minh International University, Vietnam National University. Fresh human whole blood was collected from three informed consented volunteers. The collected whole blood was immediately added to a tube containing 3.2% *w*/*v* tri-sodium citrate solution at a blood/anticoagulation (*v*/*v*) ratio of 9:1 to prevent blood coagulation. The blood was utilized within 2 h after collection to ensure normal coagulation behavior.

The blood clotting assay was performed according to the reported protocols with some modifications [31]. Briefly, the samples were cut into squares of 10 × 10 mm^2^ and placed on culture dishes. Similarly to the bacterial susceptibility testing, the electrospun PCL membrane and AQUACEL^®^ Ag Extra ^TM^ were served as the negative and positive controls, respectively. Next, 12 μL of re-calcified blood (containing 10% sodium citrate and 0.04 M CaCl_2_) was dropped on the samples. The culture dishes were incubated at 37 °C for 5 min for blood clotting, and then 3 mL of deionized water was gently added to the dishes to lyse the uncoagulated red blood cells. The concentration of free hemoglobin in distilled water was quantified by measuring the absorbance at a wavelength of 540 nm with a microplate reader (Varioskan TM, Thermo Scientific, Waltham, MA, USA). The absorbance of 12 μL citrated whole blood in 3 mL of DI was appointed to 100 as a control. The blood clotting index (BCI) was measured by using the following equation for various samples:(2)$$\mathrm{BCI}=(1-\frac{{\mathrm{OD}}_{\mathrm{sample}}}{{\mathrm{OD}}_{\mathrm{control}}})\times 100$$
where OD_sample_ and OD_control_ are absorbances of blood that has been in contact with samples and without samples in the water at 540 nm, respectively.

#### 2.2.6. In Vivo Therapeutic Effects on Wound Healing

The in vivo wound healing research was approved by the Institutional Review Board of the School of Biomedical Engineering, Ho Chi Minh International University, Vietnam National University. Fifteen male albino mice (Pasteur Institute in Ho Chi Minh City) weighing 40–45 g were randomly divided into control, COS3, COS6, PVP6–3, and PVP6–6 groups. The mice were fed ad libitum with commercial mouse food and fresh water. They were acclimatized for at least one week before experimentation. The experiment was conducted following the reported protocol with some modifications [32]. First, each mouse was anesthetized by the intramuscular injection of XYL-M2 (4 mg/kg, Xylazine, Arendonk, Belgium) and Zoletil ^®^ 50 (5 mg/kg, Tiletamine and Zolazepam, Carros, France). Then, a full-thickness incisional wound with a diameter of 10 mm was cut in the dorsal skin area of each mouse. A silicone splint with a 10 mm-wide round hole was adhered to the vicinity of the wound with cyanoacrylate adhesive. The wounds of the treated groups were covered with different sterilized samples: COS3, COS6, PVP6–3, and PVP6–6. The wounds of mice in the control group were covered with sterilized cotton gauze (Bao Thach, Binh Duong, Vietnam). New dressings were changed daily, and the wound healing progression was monitored with a digital camera. Wound closure percentages were calculated using the following formula:(3)$$\mathrm{Wound}\;\mathrm{closure}\;\left(\%\right)=\frac{A_{0}-A_{t}}{A_{0}}\times 100\;$$
where A_0_ and A_t_ are the initial wound area and wound area measured at day t, respectively.

After day 10, all the mice were sacrificed. The newly healed tissues were extracted and fixed in 10% formalin. After being stained with hematoxylin and eosin (H&E), tissue slices were observed under an inverted microscope (Nikon Eclipse, Ti-U series, Tokyo, Japan).

#### 2.2.7. Statistical Analysis

All experiments were performed in triplicate unless specified otherwise. Statistical analysis was performed using Sigma Plot V.14.0 version (SSI, USA). Differences between samples were analyzed by one-way analysis of variance (ANOVA) followed by the Tukey–Kramer post-hoc test. Data were expressed as the mean ± standard deviation, and *p* < 0.05 was considered statistically significant.

## 3. Results

### 3.1. Effect of PVP/COS on PCL/POX Surface Morphology and PCL/POX Wettability

Figure 1A shows the morphology of electrospun PCL, PCL/POX, and PCL-PCL/POX-PVP/COS membranes. The PCL membrane (Figure 1(Ai)) was composed of unaligned fibers and interconnected pores with an average diameter of 2.29 ± 0.34 µm and 8.66 ± 1.69 µm, respectively; there was no sign of bead formation. PCL/POX (Figure 1(Aii)) had a similar morphology with an average fiber diameter of 1.38 ± 0.16 µm and a pore size of 5.28 ± 1.45 µm. The coating of PVP/COS on the PCL/POX layer with different PVP concentrations and spraying times was shown in Figure 1(Aiii). In PVP4–3, the coating layer partially covered the PCL/POX layer and individual threads could also be observed. By raising PVP concentration (PVP6–3) or spraying times (PVP4–6), the PCL/POX surface was completely filled and the fibers were indistinguishable from each other. No clear differences could be observed among samples (PVP6–3, PVP6–6, PVP8–3, and PVP8–6) as the PVP concentration and coating layer increased, albeit there were small cracks emerging due to the heat expansion of coating components. The FTIR spectra of PCL-PCL/POX-PVP/COS also displays some characteristic peaks of PVP and COS (Appendix A). The results proved that the PVP/COS mixture was successfully coated onto the PCL/POX layer.

Figure 1B shows the corresponding WCA values of the PCL, PCL/POX, and PVP/COS surfaces. The PCL surface was hydrophobic with a WCA value of 119°. It was revealed that the combination of PCL and POX could improve the hydrophilicity of the PCL membrane. After 0’12”, the water droplet was completely absorbed by the PCL/POX layer, indicating super hydrophilicity. The PVP/COS coating also exhibited wettability with WCA values of PVP4–3, PVP4–6, PVP6–3, PVP6–6, PVP8–3, and PVP8–6 being 36°, 33°, 36°, 38°, 41°, and 34°, respectively. The results showed that both spraying times and PVP concentration did not cause a significant change in WCA (*p* > 0.05).

### 3.2. Water Absorption

The membranes’ ability to absorb body exudate was determined via the swelling test in simulated body serum, and the results are presented in Figure 1C. There was no difference in the water absorption capacity between COS3 and COS6, which was 2.1 and 2.9 mg/cm^2^, respectively. As PVP was loaded onto the coating, the membranes’ water retention was significantly enhanced (*p* < 0.05). The water uptake of PVP4–3 was over 7.5 mg/cm^2^, and that of PVP6–3 was 11.6 mg/cm^2^. However, the water absorbability plummeted to 6.8 mg/cm^2^ even though the concentration of PVP was raised from 6% to 8%. Surprisingly, the more layers of PVP/COS coated led to a remarkable drop in the water uptake values, being 4.2, 2.5, and 4.6 for those of PVP4–6, PVP6–6, and PVP8–6, respectively.

### 3.3. Effect of PVP/COS Coating on Mechanical Properties of Electrospun PCL/POX Membrane

The mechanical properties of PCL-PCL/POX-PVP/COS membranes were investigated via their stress-strain curves, displayed in Figure 1D. The strength at break and elongation at break of raw PCL were 3.0 MPa and 690.9%, respectively. Overall, the PCL-PCL/POX-PVP/COS membranes had better tensile strength but a lower stretching ratio compared to those of the neat PCL membrane. Furthermore, their stabilities were improved with the increase in PVP concentration. In particular, the strength at break and elongation at break of PVP4–3 were 3.8 MPa and 231.1%, while those of PVP6–3 were 4.4 MPa and 535.9%. There was no difference in the tensile properties of PVP8–3 (4.2 MPa and 551.7%) compared with PVP6–3. However, as the spraying time increased, the membranes lost their tensile-bearing capacity. For instance, at the rupture points, the strength and elongation values of PVP6–6 and PVP8–6 were 2.1 MPa-220.6% and 2.2 MPa-229.2%, respectively, roughly half of those of PVP6–3 and PVP8–3. The tensile resistance of PVP4–6 was also reduced with the ultimate strength and elongation of 3.0 MPa and 191.2%.

### 3.4. Antibacterial Activities of Membranes

The antibacterial properties of membranes were investigated by using the antimicrobial fabric zone of inhibition test (AATCC 147). Figure 2A,B present the effect of multi-sprayed membranes on two strains of bacteria, *P. aeruginosa* (negative) and *S. aureus* (positive), according to their inhibitory zones. The larger zone of inhibition indicated higher inhibitory effects of membranes on pathogens. The results suggested that these effects were PVP- and layer coating-dependent, meaning that the increase in PVP concentration and spraying times (increase PVP loaded) induced stronger antibacterial activities. In particular, COS itself hardly prevented bacterial growth. Once PVP was incorporated into the coating layer, the antibacterial effect was significantly improved with the inhibition zone diameters increased from 14.9 ± 1.4 to 20.0 ± 2.5 to 26.0 ± 2.1 mm (against PA) and from 14.8 ± 0.3 to 17.9 ± 2.5 to 25.5 ± 1.7 mm (against SA) pertaining to PVP4–3, PVP6–3, and PVP8–3. Moreover, the inhibition zone continued to expand as more PVP/COS layers were introduced. The diameters further increased from 35.7 ± 1.3 and 30.4 ± 2.6 (PVP4–6) to 39.8 ± 0.5 and 38.2 ± 1.3 (PVP6–6), and finally reached the values of 41.3 ± 0.7 and 44.3 ± 1.2 mm (against PA and SA, respectively) with the highest amount of PVP/COS in PVP8–6. The commercial wound dressing AQUACEL^®^ Ag Extra^TM^ exhibited the inhibitory zones of 13.3 ± 0.6 mm (against PA) and 14.0 ± 1.1 mm (against SA), which were as wide as those of the PVP4–3.

### 3.5. Whole Blood Clotting Assay

Figure 3 represents the blood clotting index of blood after being exposed to different membranes’ surfaces. It was clear that PVP showed a remarkable effect on blood coagulation when the PVP-contained samples had a higher impact than the non-PVP samples. However, with the higher PVP amount and coating times, the less effective blood coagulation was. Yet, the differences among PVP-contained samples were insignificant. Among the PVP-contained samples, the most and least effective clotting effects belonged to PVP4–3 and PVP8–6, which acquired 80% and 65% coagulated blood, respectively. In addition, the clotting activity was also significantly improved by the increase in COS amount in the non-PVP specimens, which had been shown in the samples of COS3 and COS6, when the former BCI value was 10 and the latter 32%. With no significant difference in blood clotting index, the hemostasis of PVP4–3 was comparable to that of the commercial wound dressing, AQUACEL^®^ Ag Extra^TM^.

### 3.6. In Vivo Therapeutic Effects on Wound Healing

Aside from the hemostatic and antibacterial effects, an ideal wound dressing should not exert any toxicity on the biological systems. An in vivo wound healing model was employed to investigate the biocompatibility of the membranes. The development of surgery sites in murine models is observed in Figure 4A, and the closure rates of all samples are presented in Figure 4B. Overall, the healing progression in all five groups was clearly observed after day 4. The wound bed slowly dried out, and the scab formation started on day 5 (Appendix A). There was no sign of bleeding or damage on the wound bed after the membrane removal, which demonstrated the antiadhesive characteristics of the coating layers. The mice of the PVP6–3 group had the fastest healing rate, while those of the COS6 and control groups were the slowest. At the end of the in vivo experiment, most of the wounds treated with COS3, PVP6–3, and PVP6–6 were recovered (the PVP6–3 group was completely healed), and the scabs fell off. On the other hand, the groups treated with COS6 and the control group had 60% of the wound area closed with the scabs still attached. The results indicated that COS3, PVP6–3, and PVP6–6 facilitated the re-epithelialization, whereas COS6 impeded wound recovery.

At the end of the in vivo wound healing experiment, all mice were sacrificed, and the newly-grown tissues were harvested and stained with H&E for histological assessment (Figure 5). Two locations, including the wounded area and the healed area, were magnified to observe the composition and structure of the regenerated tissues. Stained collagen fibers appear in pale pink, cytoplasm in dark purple, and erythrocytes in cherry red. Indications of the healing process such as blood vessels, hair follicles, and adipocytes are marked by red arrows, blue arrows, and green arrows, respectively. In the control group (Figure 5i), the new tissue was not fully reconstructed yet. The wound area had a low number of fibroblasts migrating from the edge of the wound to the central location, and only a thin layer of ECM was synthesized. Large blood vessels and growing hair follicles were observed in the hypodermis. Since the wound had not healed yet, there was no fully recovered skin area available for histological analysis. Abnormal wound healing was observed in the COS6-treated group (Figure 5iii), which was similar to that of the control group. The structure of the wound area was composed of only a thin epidermis and subcutaneous tissue, which were covered by a large scab. Enlarged photographs (Appendix A) showed the formation of blood vessels in the center spreading to both lateral areas. Granulation tissue and hair follicles were formed slowly on the far side of the wound. It can be concluded that both samples did not produce an ideal environment for skin regeneration. The lack of new blood vessels transporting nutrients and poor migration of fibroblasts to wound sites delayed tissue reparation. On the other hand, extracted tissues of mice from COS3, PVP6–3, and PVP6–6 groups were in the granulation stage with a robust proliferation of fibroblasts and collagen synthesis. In terms of the COS3 group (Figure 5ii), granulation tissue was completely established at the surgery site and was covered by a thick epidermis layer. Magnified images of granulation tissue showed a dense population of fibroblasts and a built-up extracellular matrix (Appendix A). Angiogenic capillaries were also scattered on the wound surface as well as the wound edge. Groups of hair follicles were only observed in the recovered area. The histological section of the PVP6–3 group (Figure 5iv) displayed a structure more reminiscent of that of normal skin, composed of three distinct layers: the epidermis, dermis, and hypodermis. Compared to extracted tissue from COS3 groups, the ECM was more compact, and the fibroblasts were scarcer. Adipocytes began to penetrate from the hypodermis to the dermis layer. The growth of blood vessels was noticed across the implanted area. The extracted skin treated with the PVP6–6 sample (Figure 5v) shared similar traits as those of PVP6–3. The wound was filled with thick layers of connective tissue. Newly formed capillaries and migration of adipose cells were noticed in the injury site. However, the cell population in the granulation was high, and the ECM was loosely connected.

## 4. Discussion

This study focused on designing an electrospun modified-PCL membrane coated with varying concentrations of PVP/COS. The prepared membranes were characterized and evaluated for antibacterial wound dressing applications. The addition of PVP to the COS coating layer was expected to enhance the antibacterial activity, hemostatic properties, and biocompatibility to facilitate wound healing.

Since the PCL membrane was inherently hydrophobic, the PVP/COS cannot stably attach to the membrane surface. Thus, POX, an amphiphile polymer possessing one hydrophobic block of polypropylene glycol and two hydrophilic blocks of polyethylene glycol, was mixed with PCL to alter its wettability from hydrophobicity to hydrophilicity, which proved to be successful (Figure 1B). Thanks to POX in the intermediate layer, PVP/COS was able to interact with the PCL layer and form an integrated system. From the WCA results, it was shown that the trilayer membrane PCL-PCL/POX-PVP/COS possessed an asymmetric wettability, in which the uncoated PCL layer was hydrophobic, and the PCL/POX and PVP/COS coating layers were hydrophilic. This characteristic is beneficial in practical cases where the PVP/COS side is applied directly onto the wound to absorb body exudates, while the PCL side is directed outward to repel droplets containing pathogens or other contaminants from entering the injured site.

For many daily activities, human skin is frequently under tension caused by muscle contraction and external forces. It has been reported that the tensile strength of healthy skin is around 1.8 MPa [33]. Therefore, it was crucial for an applicable wound dressing to sustain that same amount of force without rupture. From Figure 1D, it was concluded that by raising the portion of PVP in the coating solution, the mechanical properties of the membranes were significantly improved. This might be the synergistic effect between COS and PVP via hydrogen bonding, strengthening the composite structure [34]. However, high spraying times induced the opposite effect and lowered the rupture point. More likely, adding more coating layers severed the connection between each layer during the spraying and drying process and weakened the structure. Nevertheless, all samples had sufficient mechanical stability required for wound dressing.

Once the skin is injured, our body excretes fluids containing cells, electrolytes, nutrients, and growth factors that heal wounds [35]. If the wound bed is too damp, it will consequently delay the skin regeneration through the maceration mechanism [36]. To address this issue, PVP and COS were introduced onto the PCL/POX surface to improve the hygroscopicity of the membranes. The addition of PVP to the systems remarkably improved the hygroscopicity of the membranes compared to non-PVP ones (Figure 1C), which is understandable since PVP is highly hygroscopic thanks to repeat units of vinylpyrrolidone [37]. The insignificant result between PVP6–3 and PVP8–3, despite higher PVP content, might be due to the higher cross-link density between the two polymers, restricting the swelling of the coating layer [38]. On the other hand, the increase in spraying times had the opposite effect on the water retention capacity of the membranes. As aforementioned, the preparation processes impaired the integration of the systems. Thus, as the membranes were immersed in fluids, the coating layer quickly dissolved, leading to a fall in water absorbability values. In actual application, where the membranes contact the wound exudates, the coating will swell up from fluid absorption and cover the wound bed. Thanks to the loose attachment between layers, it is easy to detach PCL from PVP/COS, which will be washed off by saline solution later. In short, the dressing removal is performed with ease, and post-trauma damage is avoided.

Pathogenic bacteria could invade our body through open wounds, quickly multiply, and trigger inflammatory mechanisms. The infection prolongs the wound healing time, and it could lead to serious complications such as sepsis, organ dysfunction, and even death [39]. To address the issue, wound dressings are loaded with antimicrobial agents such as antibiotics (ciprofloxacin, gentamicin, sulfadiazine) or metallic nanoparticles (zinc oxide, silver, gold) [40]. Despite their effectiveness, these approaches still have significant drawbacks hampering their usages, such as multidrug resistance due to overuse of antibiotics [41] or harmful effects of heavy metal ions on the human body [42]. In this research, COS, a naturally-derived antibacterial product of crustacean shells, was employed. Its ability to suppress bacteria growth has been reported in many studies [43]. Although the antibacterial mechanisms of COS have yet to be elucidated, many hypotheses have been proposed [44]. Protonated amine groups on COS chains can electrically interact with the negatively charged cell membrane, which causes the leakage of intracellular constituents or disrupts nutrient transportation. On the other hand, thanks to their low molecular weight, COS molecules have better mobility and ionic interaction; hence, they can easily bind to the cell wall, obstructing microorganism growth and multiplication [45]. The agar diffusion test was conducted to assess the effect of coating layers on two bacterial strains, as demonstrated in Figure 2A,B. The results showed that at six coating times, COS with a concentration of 3% had no inhibitory effect on PA and created 14.6 mm-wide inhibitory zones against SA. Although having no effect on the microorganisms [46], the introduction of PVP to the COS coating greatly influenced the antibacterial activity of the system. PVP4–3 could create zones of inhibition with diameters of 14.9 mm and 14.8 mm against PA and SA, respectively, while those of PVP8–6 (maximum PVP concentration and spraying times) were 41.3 mm and 44.3 mm. Marina et al. showed that the presence of water-soluble PVP facilitated wetting and water penetration into the coating matrices, which led to accelerated dissolution and diffusion of drugs [47]. Likewise, the greater antimicrobial effect could be the result of improved COS release by PVP. On the other hand, with higher spraying time, the bacterial inhibition property of membranes was enhanced in relation to the increase of COS.

For hemostatic action, the higher content of COS induced a greater effect on blood clotting, as the BCI of COS6 was twice that of COS3. The BCI values further experienced a 2.5-fold increase thanks to the incorporation of PVP. The surge in BCI was likely due to the rapid release of COS, aided by PVP, as aforementioned. However, raising the PVP concentration or coating layer did not enhance the hemostatic effect of the membrane, which was consonant with a previous study [48]. Chitosan and its derivatives were reported to control the bleeding via several mechanisms. For example, positively charged amine groups can bind to the negatively charged surface of red blood cells (RBCs) and encourage RBC aggregation [49]. Chitosan promotes platelet adhesion and activation by absorbing plasma protein and fibrinogen, stimulating the formation of thrombin followed by enhancing the expression of glycoprotein IIb/IIIa, etc. [50,51,52]. COS has a relatively shorter polymer backbone than its parental chitosan due to chain scission. Therefore, its hemostatic activity declines in accordance with the lack of charge density [53]. Yet, RBCs quickly aggregated and formed dense blood clots while in contact with the PVP/COS coated surface. The phenomenon was likely the result of the COS release in bulk under the support of PVP.

The biocompatibility of the membranes was evaluated by employing the murine wound healing model. Full-thickness incisional wounds with silicone splints applied around them were created. The surgery sites were treated with samples, which were replaced on a daily basis. Wound progression was illustrated in Figure 4. The purpose of splinting was to prevent the rapid contraction of mouse skin; thus, the skin regeneration became dependent on epithelialization, cellular proliferation, and angiogenesis, which better mimicked the human wound healing process [54]. From Figure 4A, it could be seen that the wound surface was kept dry thanks to the dressing absorbing excessive fluids. The bleeding did not occur during the dressing removal thanks to the antiadhesive properties of the COS coating. Overall, samples with a low amount of COS and PVP promoted the re-epithelialization and accelerated the healing rate of the wound, especially with PVP6–3. A high dose of COS released from COS6 samples might establish an unfavorable environment for skin regeneration [32]. On the other hand, PVP6–6, despite having an equal amount of COS to COS6, facilitated the process as much as COS3 and PVP6–3 did. It could be explained that due to the electron-rich oxygen atoms, the C = O groups of PVP could act as proton-acceptors, interacting with the protonated amine groups of COS. Thanks to part of amine groups being linked with PVP, COS then had a lower impact on the tissue. A similar effect of PVP enhancing the cytocompatibility of COS was also reported previously [55].

The histological study was conducted to confirm whether any defection occurred in the new growth tissue. Since the contraction was greatly obstructed by splints, the wound healing mechanism relied mostly on the migration of fibroblasts from the edge of the wound to the center, their proliferation, and the reconstruction of new ECM to fill the missing areas. Overall, on day 10 of the experiment, the inflammation had stopped, and the epithelium layers had merged to re-epithelialize the wound in all treated groups. H&E staining images of COS3, PVP6–3, and PVP6–6 confirmed the formation of granulation tissue. In the cases of COS3 and PVP6–6, the fibroblast density was high, and the newly formed ECM was loosely connected. On the other hand, the compacted ECM with a low fibroblast population observed in PVP6–3 was closely related to normal wound healing, in which granulation tissue begins to disappear at the time when re-epithelialization finishes through massive apoptosis of fibroblasts and vascular cells [56]. Extracted tissue from COS6 groups had the slowest healing rate among the treated samples. There was a large scab covering the wounded area, and thin granulation tissue was supported by a robust vascular network. The result reconfirmed that a high amount of COS could produce an unfavorable environment for tissue regeneration.

In this study, PVP6–3 was the most promising candidate for wound dressing application thanks to its excellent antibacterial activity, hemostatic action, and biocompatibility. COS3 showed good in vivo biocompatible results but was unable to induce an inhibitory zone as well as a blood clotting effect. On the other hand, COS6 samples showed weak performance against bacteria and produced irregular tissue in the murine model. Overall, this study has demonstrated a simple approach of combining COS bioactive agent onto the electrospun PCL membrane via surface modification with POX and coating techniques. The synergistic effect of COS and PVP was exploited to improve the properties of the PCL-PCL/POX-PVP/COS membrane in terms of mechanical strength, hygroscopicity, antibacterial and hemostatic activities, and biocompatibility.

The PVP/COS combination could be developed into a versatile coating and introduced to other membranes to fabricate different wound dressings for different wound treatments. However, there are technical challenges that should be considered in future works. The spraying process was performed manually, offering restricted control (spraying time and distance between spraying nozzle and membrane) over the PVP/COS thickness distribution. Mistargeting and over-spraying without proper drying time were responsible for the loss of PVP/COS solution. The drying stages between each spraying time were time-consuming, and the prolonged exposure to high temperatures could damage the membrane integrity. To address the issues, we suggest reducing the coating volume in each cycle while increasing the spraying time. Using a warm air blower instead of an oven could reduce the drying time without damaging the membrane structure. For future prospects, the PCL-PCL/POX-PVP/COS membrane can be tested on other in vivo wound models such as infectious, diabetic, or burn models, given their hygroscopicity, antibacterial activity, and therapeutic efficacy.

## 5. Conclusions

The study has reported the effect of the PVP/COS coating on the electrospun modified-PCL membrane for wound dressing application. The PCL-PCL/POX-PVP/COS possessed asymmetric wettability thanks to the trilayer design composed of the hydrophobic PCL (WCA value of 119°) and the hydrophilic PVP/COS (WCA values of 33–41°). The increase in PVP concentration enhanced the mechanical stability and hygroscopicity, whereas the increase in spraying times dramatically degraded both characteristics. The incorporation of PVP into the systems also greatly improved the antibacterial and hemostatic activity of the membrane. In terms of wound healing promotion, COS3, PVP6–3, and PVP6–6 showed good in vivo biocompatibility while accelerating skin regeneration. In conclusion, PVP6–3 was the optimal membrane with suitable physicochemical and biological characteristics. The sample can absorb the amount of fluid of 11.6 mg/cm^2^ and withstand the maximum tensile strength of 4.4 MPa. PVP6–3 produced zones of inhibition of 20.0 ± 2.5 and 17.9 ± 2.5 mm against PA and SA, respectively, while exhibiting hemostasis with BCI of 74%. Furthermore, mice treated with PVP6–3 reached 95% wound closure on day 10, and the harvested skin structure was similar to that of native tissue. The simple approach demonstrated here could be applied to the future fabrication of a multi-purpose wound dressing.

## Data Availability

The data used to support the findings of this study are included in the article and Appendix A.

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
