# Peer review of "Optimization of Oligomer Chitosan/Polyvinylpyrrolidone Coating for Enhancing Antibacterial, Hemostatic Effects and Biocompatibility of Nanofibrous Wound Dressing"

_polymers, 2022, doi:10.3390/polym14173541_

Round 1

Reviewer 1 Report

This manuscript reported trilayer dressings with asymmetric wettability, composed of electrospun polycaprolactone (PCL) base membranes coated with oligomer chitosan (COS) in various concentrations of polyvinylpyrrolidone (PVP), and they are fabricated for wound dressing application. The work is abundant and comprehensive and it’s worthy to publish. However, the manuscript needs some modifications so that it could be better than before. It would be helpful if authors would consider about the following points:

1.      The section “abstract” is suggested to add more specific data to enhance the reader's impression for this work, rather than write some subjective evaluation.

2.      The section “2.2 Methodology” is complicated, it is encouraged to add a flow chart of preparation in this section.

3.      In Table 1, why do you choose these amounts in this experiment (e.g., 3% for COS concentration, 0-8% for PVP concentration, 3 and 6 for spraying times)? Please explain it or add some relevant references.

4.      The design of figures should be updated. For example, Fig.1 is irregular.

5.      The function of discussion and that of conclusion are repetitive. Some specific data and concise results should be existent in the conclusion. Thus, the conclusion should be re-written.

6.      It is encouraged to state the main limitations of this study and present some suggestions for future researches.

7.      The English level throughout the manuscript should be improved to meet the standard.

8.      There are some errors of formats and grammars, please check the paper carefully. For instance, H2O2 (line 113) should be corrected, there are also some errors in line 161-164, etc.

9.      The references nearly 5 years are very few, and the central references are too old.

10.  The format of references should be updated, there are some errors, please check the paper carefully.

Author Response

First and foremost, we would like to express our gratitude for your kind suggestions for our manuscript. Here are our responses to your review report:

  1. We gave some data in the abstract and reduced the subjective statements.
  2. A graphical abstract presenting the fabrication process was added.
  3. The COS solution in this research was developed based on our previous study:'' Characterizations and Antibacterial Efficacy of Chitosan Oligomers Synthesized by Microwave-Assisted Hydrogen Peroxide Oxidative Depolymerization Method for Infectious Wound Applications" (https://doi.org/10.3390/ma14164475). In this study, COS 3% was synthesized by depolymerization of chitosan, and its antibacterial activity was evaluated. We noticed that although chitosan is known for its antibacterial activity, obtained COS did not show such property. Furthermore, COS has low water absorbability and is weak in terms of mechanical strength. Hence, the PVP was added to the COS coating to improve the overall properties of the system. In many studies, the PVP:COS ratios were often fixed, so the understanding of PVP influence on the overall properties is still lacking. Therefore, we varied the PVP/COS ratios and spraying times to bring out the PVP and COS potential.    
  4. We divided figure 2 and figure 3 into smaller parts for better visualization, as another reviewer suggested. However, regarding figure 1, we are uncertain of main problems. We hope that you could give us more details about the issue that you found irregular so that we will fix it in the next revision
  5. The conclusion was revised, and some important data were mentioned.
  6. We described some challenges we experienced during our research and gave some suggestions to address the issues. It involved the manual spraying step, which offered poor control over the membrane thickness and loss of coating solution during the process. Also, the drying stage using the oven was time-consuming, not to mention the higher temperature could damage the membrane. Our suggestion is to reduce the coating volume each cycle while increasing the cycle times for better PVP/COS solution absorption, and using warm air blower to improve the drying process. 
  7. We revised the entire manuscript again with the help of Grammarly to correct the grammatical mistakes.
  8. We corrected the errors.
  9. We replaced old references with new ones.
  10. We are sorry for the mistakes. We used the cross-reference for automatically updating the figure numbering. When we moved the manuscript content to the journal template, the cross-reference malfunctioned, so the figure number could not be showed. We reviewed the whole manuscript and corrected these errors.

We hope our reply meet your expectation. Thank you for taking your time for us.

Reviewer 2 Report

The study is on the preparation of polycaprolactone (PCL)- based polyvinyl pyrrolidone (PVP) and oligomer chitosan (COS)-coated tri-layer PCL-PCL/POX-PVP/COS composite fibrous membranes through electrospinning and spraying techniques for wound healing applications. Results of the study showed that the wound dressing coated by spraying 3 times the solution of 3% COS and 6% PVP on the PCL base membrane exhibits the highest performance in improving in vivo wound healing.

The study is well-intended and well-organized containing comprehensive characterizations. There are a few items that need to be corrected and revised before publication. Specific comments are listed below:

1.     It is difficult to follow the designation of the samples such as the code PCL-PCL/POX-PVP/COS and then followed by COS3, PVP 8-3, PVP8-6… etc. Therefore, the last paragraph of the introduction section needs to be revised to avoid possible confusion. E.g. writing the code of the overall trilayer system as PCL-PCL/POX-PVP/POX. I think it is supposed to be PCL-PCL/POX-PVP/COS.   

2.     The manuscript needs to be proofread for typological errors. Through the manuscript instead of the Figure number, it is written “Error! Reference source not found”.

3.     Mechanical characterization details should be given (sample size and shape, 5x 5 cm ? loading speed).

4.     In Fig. 2 it is difficult to follow the inset images. It is recommended to give them separately or revise the dimensions.

5.     Above comment is also valid for Fig.3 especially (B). It is recommended to present Fig 3-B separately.

6.     The relevant permit numbers for the in vitro blood coagulation and the in vivo animal experiments should be provided. 

7.      It may be useful to measure the membrane thickness as well as the PVP/COS coating layer thickness.

8.     Fiber and pore diameter measurements should be described in detail. If it is based on SEM images the number of selected fibers should be provided.

Author Response

First and foremost, we thank you for your kind suggestions for our manuscript. Here are our responses to your review report:

  1. To address the issue of the membranes' abbreviation, we decided to modify table 1. Now, table 1 gives out the information about the membranes, including the abbreviation of each layer and their compositions. We also made a minor correction at the overall membrane (PCL-PCL/POX-PVP/COS), just as you have pointed out.
  2. We are terribly sorry for the mistakes. We used the cross-reference for automatically updating the figure numbering. When we moved the manuscript content to the journal template, the cross-reference malfunctioned, so the figure number could not be showed. We reviewed the whole manuscript and corrected these errors. 
  3. Details of the tensile test were added to the manuscript, including the sample shape (rectangular), size (40x10 mm2, and loading speed (8.3 mm/second).
  4. As your suggestion, we separated figure 2 into two graphs for better observation.  
  5. As your suggestion, we separated figure 3 into two graphs for better observation.  
  6. The in vitro blood coagulation and the in vivo animal experiments were approved by the Institutional Review Board of the School of Biomedical Engineering, Ho Chi Minh International University, Vietnam National University. However, there were no study/project numbers assigned, so we cannot provide them.
  7. We do not have the necessary technique at the moment for the measurement of membrane thickness. We thought of using AFM method for the measurement. Since we do not have AFM, the only option is to ask for the out-of-school service, which requires a lot of time and money.
  8. We provided the details of fiber and pore size measurement to the manuscript. Based on the SEM micrographs, we randomly measured 30 fibers and 30 pores using ImageJ software, and reported the average values.

We hope our reply meet your expectation. Thank you for taking your time for us.

Reviewer 3 Report

This manuscript demonstrates a multilayer membrane design for an ideal wound dressing. Polycaprolactone (PCL) is used as structural support, whereas oligomer chitosan (COS) in various concentrations of polyvinylpyrrolidone (PVP) is used for hygroscopicity, antibacterial, hemostatic activities, and biocompatibility. In terms of applications, they looked at antibacterial activity, blood coagulation, and in vivo wound healing. The results presented here are promising. However, there are several major flaws in the manuscript that need serious attention from the author.  

1.     First of all the referencing need a major correction. More than half of the references can’t be accessible. Instead of reference it was showing “Error! Reference source not found”. This kind of mistake is not acceptable. The introduction and conclusion need re-writing.

2.     The use of PVP-Chitosan in wound healing is not new. It is frequent in literature. The author has not discussed enough them in the introduction. Consequently, the novelty of this work is not coming out anywhere.

3.    In the fabrication of PCL and PCL/POX bilayer membrane the author mentioned that the acetone solution of the polymer mixtures was stirred overnight at 50 oC. Just curious to know how acetone survives at 50 oC overnight.

4.     In antibacterial activity testing the author has not used any positive control like PEI or poly-Lys etc. Without any standard +ve control it is hard to rationalize how superior these membranes are. In addition, reporting antibacterial activity just by reporting areas of a clear zone is a casual way of reporting. It should be compared with the standard sample.  

5.     Same thing as above should be followed for blood clotting assay.

6.     There are several errors in superscript and subscript.

Author Response

First and foremost, we thank you for your kind suggestions for our manuscript. Here are our responses to your review report:

  1. We are terribly sorry for the mistakes. We used the cross-reference for automatically updating the figure numbering. When we moved the manuscript content to the journal template, the cross-reference malfunctioned so the figure number could not be showed. We had reviewed the whole manuscript and corrected these errors.
  2. The COS solution in this research was developed based on our previous study:'' Characterizations and Antibacterial Efficacy of Chitosan Oligomers Synthesized by Microwave-Assisted Hydrogen Peroxide Oxidative Depolymerization Method for Infectious Wound Applications" (https://doi.org/10.3390/ma14164475). In this study, COS 3% was synthesized by depolymerization of chitosan, and its antibacterial activity was evaluated. We noticed that although chitosan is known for its antibacterial activity, obtained COS did not show such property. Furthermore, COS has low water absorbability and is weak in terms of mechanical strength. Hence, the PVP was added to the COS coating to improve the overall properties of the system. In many studies, the antibacterial, hemostatic, and biocompatible properties of the composite were enhanced by the addition of other materials such as graphene oxide or bentonite. In stead, our study utilized the synergistic effect of PVP and COS itself to perform such functions. Furthermore, The PVP:COS ratios were often fixed, so the understanding of PVP influence on the overall properties is still lacking. Therefore, we varied the PVP/COS ratios and spraying times to bring out the PVP and COS potential.     
  3. Since the acetone solvent easily evaporates even at the ambient condition. The dissolution of PCL using acetone at 50oC was conducted in a capped vial. To prevent any leaking of acetone from the vial, we also seal the gap between the cap and the vial body with plastic wrap. We marked the level of solvent outside the vial to check for the drop of volume. After 24 hours, We noticed that the solvent level inside the vial was unchanged.
  4. We had conducted and reported the results of the AATCC 147 test on the negative and positive control, which were raw PCL and AQUACEL wound dressing.
  5. We had conducted and reported the results of the Whole blood test on the negative and positive control, which were raw PCL and AQUACEL wound dressing.
  6. We reviewed the manuscript and corrected the mistakes.

We hope our reply meet your expectation. Thank you for taking your time for us.

Round 2

Reviewer 1 Report

This manuscript reported trilayer dressings with asymmetric wettability, composed of electrospun polycaprolactone (PCL) base membranes coated with oligomer chitosan (COS) in various concentrations of polyvinylpyrrolidone (PVP), and they are fabricated for wound dressing application. The manuscript was well organized after modifying. I recommend to accept it for publication.

Reviewer 3 Report

The paper can be accepted in its present form.